# The First Complete Mitochondrial Genomes of Two Sibling Species from Nitidulid Beetles Pests

**DOI:** 10.3390/insects11010024

**Published:** 2019-12-28

**Authors:** Yi Wu, Yangming Lan, Liyuan Xia, Miao Cui, Weiwei Sun, Zhen Dong, Yang Cao

**Affiliations:** 1Academy of National Food and Strategic Reserves Administration, No. 11 Baiwanzhuang Street, Beijing 100037, China; zndlym@163.com (Y.L.); xly@chinagrain.org (L.X.); cm@chinagrain.org (M.C.); sww@chinagrain.org (W.S.); dongz@chinagrain.org (Z.D.); cy@chinagrain.org (Y.C.); 2Department of Entomology, College of Plant Protection, China Agricultural University, Beijing 100193, China

**Keywords:** storage insect, nitidulid beetles, mitochondrial genome, evolution

## Abstract

*Carpophilus dimidiatus* (Fabricius, 1792) and *Carpophilus pilosellus* Motschulsky are two sibling species and economically important storage pests worldwide. The first complete mitochondrial (mt) genomes of both were sequenced using next-generation sequencing. The mt genomes of *C. dimidiatus* and *C. pilosellus* are circular, with total lengths of 15,717 bp and 15,686 bp, respectively. Gene order and content for both species are similar to what has been observed in ancestral insects and consist of 13 protein-coding genes, two ribosomal RNA genes, 22 transfer RNA genes, and a control region. Comparing the mt genome data of *C. dimidiatus* and *C. pilosellus*, they are similar in organization, arrangement patterns, GC contents, transfer RNA (tRNA) secondary structures, and codon usage patterns. Small differences were noted with regards to the nucleotide similarity of coding regions and the control region. This is the first publication of the complete mitochondrial genomes of two sibling species. The mt genome sequences can supplement the nuclear markers of the *Carpophilus* genus in research species identification, system evolution, and population genetic structure, and also will be valuable molecular marker for further genetic studies.

## 1. Introduction

The beetle genus *Carpophilus* (Coleoptera: Nitidulidae: Carpophilinae) is one cluster in taxonomic lineage. There are about 200 *Carpophilus* species, most of which are distributed in the tropics and subtropics [1,2,3,4]. Some species are considered important stored-product pests. *Carpophilus dimidiatus* (Fabricius, 1792) and *Carpophilus pilosellus* Motschulsky are two sibling species that show a strong preference for attacking stored grain, dry fruits, Chinese herbal medicine, and many other stored-product commodities. They reduce the quality and quantity of stored products and create conditions for the growth of molds and fungi [5,6,7]. *Carpophilus dimidiatus* threatens the brewing and fermentation industry, and is distributed worldwide [8,9]. *Carpophilus pilosellus* is often found in dry fruit storage sites and has a received widespread distribution across China, Japan, Vietnam, and India [10,11].

Very few studies focused on both *C. dimidiatus* and *C*. *pilosellus* have been reported, except for harm, taxonomy, and first records, in some regions and countries [1,10]. Molecular information on the two species is inadequate, with only few reports about the molecular identification *C. dimidiatus* by species-specific PCR and barcode fragments of mitochondrial cytochrome c oxidase I (*cox*1) gene [8,9]. The molecular data of storage pests are very useful, not only to study the population structure and geographical variability but also to resolve ecological and taxonomic relationships [12,13,14].

In most insects, mitochondrial (mt) genomes contain a circular chromosome and have a relatively stable structure. The mt genomes share an ancestral pattern of gene arrangement, containing 37 genes: 13 protein-coding genes (PCGs), 22 transfer RNA (tRNA) genes, and two ribosomal RNA (rRNA) genes [15,16,17]. The whole mt genome is playing an increasingly important role in intraspecific and interspecific genetic differences, phylogeographic studies, and rapid diagnostics [18,19,20]. At the time of this report, no mt genomes of *Carpophilus* species were reported in GenBank. In this study, we report the first complete mt genomes of *C. dimidiatus* and *C. pilosellus*, analyze the gene arrangement, and compare the mt genome data between the two sibling species. The results provide valuable nuclear markers for the further study of the genetic structure of populations and the phylogenetic analysis of nitidulid beetles.

## 2. Materials and Methods

### 2.1. Sampling and DNA Extraction

*Carpophilus dimidiatus* was collected from a grain processing facility at Hujian—Zhangzhou, China, in 2016, and *C. pilosellus* was collected from dry fruit storage sites at Hebei—Qinhuangdao, China, in 2017. All adult samples were identified by their morphology features [11]. Genomic DNA was extracted from three adults that been pooled into one centrifuge tube, using the TIANamp Micro DNA Kit (TIANGEN Biotech, Beijing, China) following manufacturer’s instructions. DNA concentrations were tested using a UV-Vis Spectrophotometer (Q5000, Quawell Technology, Inc., Sunnyvale, CA, USA).

### 2.2. Sequencing and Assembly

The two sequencing libraries were prepared with total DNA of *C. dimidiatus* and *C. pilosellus*, respectively. Sequencing was carried out on Illumina Hiseq2500 with 500 cycles and 250 bp paired-end sequencing at the Berry Genomics Company, Beijing, China. 

A quality assessment of raw FASTQ files for the two libraries was carried using FastQCv0.11.8 (http://www.bioinformatics.babraham.ac.uk/projects/fastqc/). The adapter sequences were removed with Trimmomatic v0.30 using the parameter LEADING:3, TRAILING:3, SLIDINGWINDOW:4:5, MINLEN:25 [21]. We applied “map to reference” strategy and mapped all cleaned reads to the “anchor”. The *cox*1, *rrnS*, and *rrnL* gene fragments were sequenced as the “anchor”. Illumina sequence-reads were assembled with Geneious11.0 [22]. This process was done iteratively over 100 times for each of the reference sequence in order to get the mt sequence. The results across the three references are identical for the both species. The contigs were assembled to the full circular mt sequences using the following parameters: (1) minimum overlap 50 bp and (2) minimum similarity 99%.

We sequenced *cox*1, *rrnS*, and *rrnL* gene fragments using universal primer pairs LCO1490–HCO2198 [23], 12SF–12SR, and 16Sar–16Sbr [24,25]. Polymerase chain reaction (PCR) amplification in final reaction volume of 25 µL included 12.5 µL of 2× Taq MasterMix (Tiangen, Beijing, China), 9.5 µL ddH_2_O, 1 µL of each primer (10 µM), and 1 µL genomic DNA. PCR cycler conditions were 95 °C or 3 min, followed by 35 cycles of 94 °C for 1 min, 50 °C for 1 min, and 72 °C for 1 min, with the final extension at 72 °C for 10 min. The amplicons were sequenced at the Berry Genomics Company, Beijing, China.

### 2.3. Gene Identification and Analyses

The protein coding genes (PCGs) and ribosome RNA (rRNA) genes were identified using the MITOS Web Server (http://mitos.bioinf.uni-leipzig.de/index.py) [26]. The “RefSeq 81 Metazoa” was selected as reference. Then, NCBI BLASTn with parameters (*e*-value < 0.001, identity > 90%) was applied to confirm the annotation results from last step. Transfer RNA (tRNA) genes were identified using tRNA-Scan [27] and ARWEN [28] based on secondary structures. If all 22 tRNAs were not found using this software, we alternatively found them by manual operation according to the codon and structure of tRNAs. MEGA 7.0 [29] was initiated to analyze the nucleotide composition, codon usage, and relative synonymous codon usage. Nucleotide compositional skew was calculated using the formula AT skew = A − T/A + T and GC skew = G − C/G + C. The circular maps of *C. dimidiatus* and *C. pilosellus* mt genome sequences were drawn with CG View40 [30]. The annotated mt genome sequence were deposited in GenBank with their accession number (submission ID were MN604384 and MN604383).

## 3. Results

### 3.1. Mt Genome Sequencing and Assembly

Both libraries of *C. dimidiatus* and *C. pilosellus* were sequenced on Hiseq2500. A total of 486,726 read-pairs for *C. dimidiatus* and 516,913 read-pairs for *C. pilosellus* were generated after removing the low-quality value reads (lower than Q20). Using the map to reference strategy with the *cox*1, *rrnS*, and *rrnL* genes as anchors, 56,246 reads were assembled for *C. dimidiatus* and 61,215 reads were assembled for *C. pilosellus*. After generating all the assembled reads, consensus sequence lengths of 16,313 bp for *C. dimidiatus* and 16,176 bp for *C. pilosellus* were generated. Finally, the repeats were detected manually to form the complete mitochondrial genome sequence.

### 3.2. Mt Genome Organization and Nucleotide Composition

The length of the mt genome of *C. dimidiatus* was 15,717 bp, and that of *C. pilosellus* was 15,686 bp. The sequenced mt genome was assembled into one circular contig, similar to many other insect mt genomes (Figure 1 and Table 1), containing 13 PCGs, 22 tRNA genes, two rRNA genes, and a non-coding region. Both of the two species share the same arrangements: four PCG genes (*nad*1, *nad*4, *nad*4*l,* and *nad*5), two rRNA genes (*rrnS* and *rrnL*), and eight tRNA genes (*trnV*, *L*1, *P*, *H*, *F*, *Y*, *C,* and *Q*) were located in the light strand, and the other genes were encoded by the heavy strand. 

The symmetric nucleotide compositions of *C. dimidiatus* are A: 39.0%, C: 14.9%, G: 9.9%, and T: 36.2%, and of *C. pilosellus* are A: 39.6%, C: 13.4%, G: 9.4%, and T: 37.6%. As in mt genomes from other insect species, the nucleotide composition of the two nitidulid beetles was biased on A and T in *C. dimidiatus* (AT: 75.2%) and *C. pilosellus* (AT: 77.2%). Similarly, the mt genomes of both *C. dimidiatus* and *C. pilosellus* had positive AT-skews (0.037 and 0.026) and negative GC-skews (−0.202 and −0.175) (Table 2). The PCGs, rRNAs, and tRNAs all had positive AT skews and negative GC skews. 

The total length of the intergenic sequences in *C. dimidiatus* was 120 bp, including one 24 bp intergenic sequence located between *trnC* gene and *trnW* gene. The mt genome of *C. pilosellus* had 190 bp intergenic sequences. Two intergenic sequences was longer than 20 bp. One was 23 bp, located between *trnV* gene and *trnL* gene, and the other was 109 bp, located between *trnC* gene and *trnW* gene. The other intergenic sequences were shorter than 20 bp. The total lengths of the overlapping regions in *C. dimidiatus* and *C. pilosellus* were 82 bp and 101 bp, respectively (Table 1). All the overlapping sequences of the two mt genomes ranged from 1 to 35 bp; the longest was between *trnK* and *cox*2.

### 3.3. Protein Coding Genes

The size of 13 PCGs of *C. pilosellus* was 11,124 bp, with 33 bp for stop codons. All PCGs could be translated into 3697 amino acid residues. Three types of start codons—ATA, ATT, and ATG—were used (Table 1). *nad*4, *nad*6, and *atp*6 genes started with ATA, while *nad*2, *nad*3, *nad*5, *cox*1, and *cox*2 started with ATT; other genes started with ATG. Additionally, there were three types of stop codon: TAG, T, and TAA. The *nad*1, *nad*3, *cob,* and *atp*8 genes stopped with TAG; the *cox*2, *cox*3, and *nad*4 genes had incomplete stop codons T; and the other genes stopped with TAA.

For *C. dimidiatus,* the total size of the 13 PCGs was 11,111 bp, which could be translated into 3692 amino acid residues. Start codons included four types: ATA, ATT, ATG, and ATC. The *cob*, *cox*3, *atp*6, and *nad*4*l* genes used ATG as the start codon, *nad*1 and *nad*6 were started with ATA, and ATC was used to start *cox*2 and *atp*8. The other genes started with ATT. For the stop codons, *nad*1, *nad*3, *atp*8, and *cob* were stopped by TAG. T was the incomplete stop codon of *nad*4 and *cox*3. Other genes ended with TAA.

Figure 2 shows a comparison of the relative synonymous codon usage (RSCU) of the two mt genomes. The two species shared the same codon families and have the similar feature of RSCU. The *trnS1* codon UCN was the most commonly used. Furthermore, UUA (*trnL*1), UCU (*trnS*1), and CGA (*trnR*) were the top three codons used in the two mt genomes.

### 3.4. The tRNA and rRNA Genes

Twenty-two typical tRNAs were found in *C*. *dimidiatus* and *C*. *pilosellus* (Figure 3). The two sibling species had similar features in the tRNA and rRNA genes. Most tRNAs could be folded into the clover-leaf secondary structures, except for *trnR*, *trnG*, *trnW,* and *trnH*, which lacked the TΨ C-loop; meanwhile, *trnS*1 lacked the dihydorouridine (DHU) arm. Moreover, *trnL*1 of *C. pilosellus* lacked the TΨ C-loop. The number of base pairs in the 22 tRNAs ranged from 62 bp (*trnL*1 and *trnC*) to 71 bp (*trnK*). The size of the DHU-stem ranged from three to four. Most of the TΨ C-stems had 5 bp, while stems for six of the tRNAs (*trnC*, *trnE*, *trnI*, *trnF*, *trnP,* and *trnT*) had 4 bp and the TΨ C-stems of *trnP* and *trnT* was 6 bp. The features of tRNAs were conserved between the two genomes that were usually observed in most of insects.

The two genes encoding the ribosomal subunits were located between *trnV* and the control region, and between *trnL*1 and *trnV*. The two rRNA genes were 1275 bp (*rrnL* gene) and 789 bp (*rrnS* gene) for *C. dimidiatus* and 1268 bp (*rrnL* gene) and 788 bp (*rrnS* gene) for *C*. *pilosellus*.

## 4. Discussion

We compared the mt genomes of *C. dimidiatus* and *C. pilosellus* regarding GC contents, tRNA secondary structures, codon usage patterns, and intron characteristics. Most of these features were similar between the two species and the gene order, and other structural features were largely conserved. The main difference was the nucleotide similarity between the sibling species (Table 2). The coding regions for both mt genome data were 15,717 bp and 15,686 bp, with a nucleotide similarity of 86.7%. PCGs had a nucleotide similarity of 86.60%, including 1706 variable positions. The *nad4l* had the highest similarity (89.36%), and *cox2* had the lowest similarity (76.63%). The average nucleotide similarity of all the *cox* genes was 83.51%, and the average similarity of the *nad* genes was 84.58%. According to the similarity, we can choose different mt genes to conduct intraspecific or interspecific studies. The mt genome of *C. dimidiatus* and *C. pilosellus* sequenced here was of a similar gene organization, order, and size to the published mt genome of Coleoptera stored grain pests, e.g., *Cryptolestes ferrugineus* (Stephens) was 15,511 bp [31] and *Tribolium castaneum* Herbst was 15,881 bp [32]. Here, the first two mt genome sequences for species within the genus *Carpophilus* were reported, and more data from other species in this genus will be needed for further research. Mt genome sequences will enable the resolution of species identification, phylogenetics studies, and molecular evolution of nitidulid beetles.

## 5. Conclusions

Through illumina DNA sequencing and assembly, the mt genome sequences of the two sibling species *C. dimidiatus* and *C. pilosellus* were obtained, and the coding regions were 15,717 bp and 15,686 bp, respectively. Except for the differences in nucleotide composition, they share similar organization and arrangement patterns. This study is the first description of the complete mitochondrial genome for *Carpophilus* species. The mt genome sequences will be important resources for further molecular studies with *C. dimidiatus* and *C. pilosellus*. 

## Figures and Tables

**Figure 1 insects-11-00024-f001:**
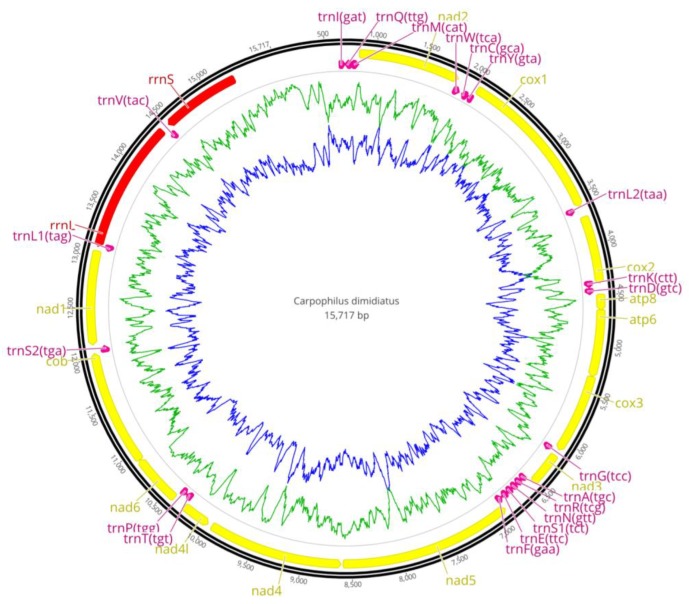
The mitochondrial genome arrangements of *C. dimidiatus* and *C. pilosellus*. The yellow is protein-coding genes (PCGs), the red is rRNA genes, and the purple is transfer RNA (tRNA) genes. The AT content and GC content were plotted using a blue and green sliding window, respectively, as the deviation from the average AT content and GC content of the entire sequence.

**Figure 2 insects-11-00024-f002:**
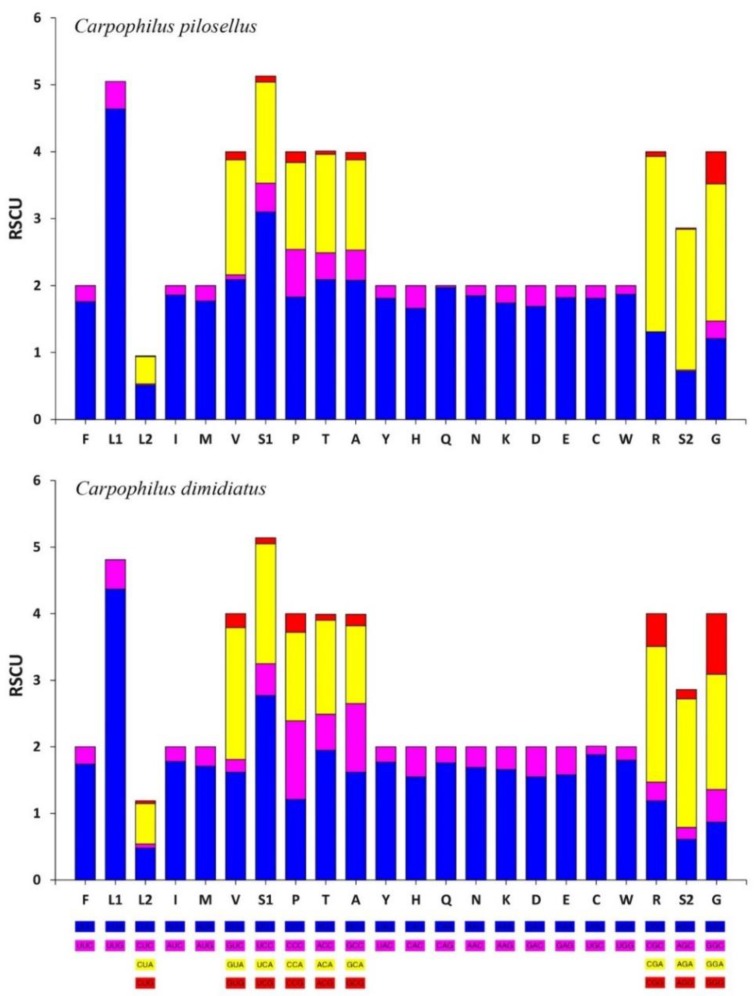
Relative synonymous codon usage (RSCU) for the mitochondrial genomes of *C. dimidiatus* and *C. pilosellus*. Codon families are shown on the X-axis.

**Figure 3 insects-11-00024-f003:**
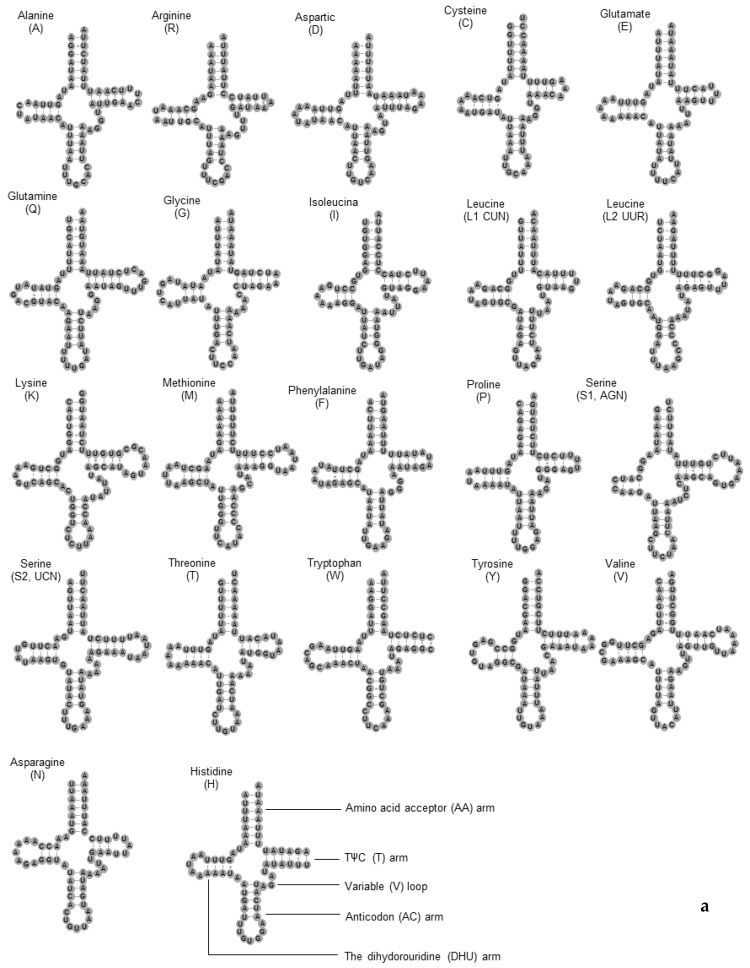
Putative secondary structures of tRNAs found in the mitochondrial genome of *C. dimidiatus* (**a**) and *C. pilosellus* (**b**).

**Table 1 insects-11-00024-t001:** Annotation of the mitochondrial genome of *Carpophilus dimidiatus* and *Carpophilus pilosellus*.

Gene	Strand	Position	Length (bp)	Anti/Start Codon	Stop Codon	Intergenic Sequence (bp)
*rrnS*	N	14505-15293/14565-15352	789/788			0
*trnV* (tac)	N	14436-14504/14496-14564	69			0
*rrnL*	N	13144-14418/13205-14472	1275/1268			15/23
*trnL*1 (tag)	N	13085-13146/13146-13207	62			1
*nad*1	N	12133-13065/12195-13142	933/948	ATA/ATG	TAG	19/3
*trnS*2 (tga)	J	12048-12115/12110-12177	68			17
*cob*	J	10910-12049/10969-12111	1140/1143	ATG	TAG	−2
*nad*6	J	10404-10910/10463-10969	507	ATA	TAA	−1
*trnP* (tgg)	N	10337-10399/10396-10458	63			4
*trnT* (tgt)	J	10273-10336/10332-10395	64			0
*nad*4*l*	N	9989-10270/10048-10329	282	ATG	TAA	2
*nad*4	N	8669-9980/8716-10051	1312/1336	ATT/ATA	T	8/−4
*trnH* (cac)	N	8608-8671/8666-8730	64/65			−3/−15
*nad*5	N	6880-8592/6938-8665	1713/1728	ATT	TAA	17/0
*trnF* (gaa)	N	6832-6896/6891-6954	65/64			−17
*trnE* (ttc)	J	6771-6833/6830-6892	63			−2
*trnS*1 (tct)	J	6704-6770/6763-6829	67			0
*trnN* (gtt)	J	6640-6703/6697-6762	64/66			0
*trnR* (tcg)	J	6577-6639/6635-6697	63			0/−1
*trnA* (tgc)	J	6513-6577/6571-6635	65			−1
*nad*3	J	6161-6514/6225-6572	354/348	ATT	TAG	−2
*trnG* (tcc)	J	6097-6160/6155-6218	64			0/6
*cox*3	J	5310-6096/5368-6154	787	ATG	T	0
*atp*6	J	4642-5310/4703-5368	669/666	ATG/ATA	TAA	−1
*atp*8	J	4493-4648/4551-4706	156	ATC	TAG	−7/−4
*trnD* (gtc)	J	4427-4492/4485-4550	66			0
*trnK* (ctt)	J	4352-4422/4410-4480	71			4
*cox*2	J	3664-4386/3722-4409	723/688	ATC/ATT	TAA/T	−35
*trnL*2 (taa)	J	3599-3663/3657-3721	65			0
*cox*1	J	2059-3603/2117-3661	1545	ATT	TAA	3/−5
*trnY* (gta)	N	2001-2066/2059-2124	66			−8
*trnC* (gca)	N	1939-2000/1997-2058	62			0
*trnW* (tca)	J	1848-1914/1821-1887	67			24/109
*nad*2	J	837-1844/828-1817	1008/990	ATT	TAA	3
*trnM (cat)*	J	768-836/741-809	69			0/18
*trnQ* (ttg)	N	696-764/672-740	69			3/0
*trnI* (gat)	J	634-698/611-674	65/64			−3

Note: N and J indicates that the gene was located in the minor (N) and major (J) strand. The ‘/‘ indicates that the left was *C. dimidiatus* and the right was *C. pilosellus*. Intergenic sequence: positive numbers/negative numbers indicate intergenic/overlapping regions between adjacent genes.

**Table 2 insects-11-00024-t002:** Nucleotide composition and similarity of mitochondrial genomes of *C*. *dimidiatus* and *C*. *pilosellus*.

Feature	A + T%		AT-Skew		GC-Skew		Similarity
	*C*. *dimidiatus*	*C*. *pilosellus*	*C*. *dimidiatus*	*C*. *pilosellus*	*C*. *dimidiatus*	*C*. *pilosellus*	
Whole mt genome	75.2	77.2	0.037	0.026	−0.202	−0.175	86.7
PCGs	74.3	76.6	0.039	0.023	−0.191	−0.149	86.6
tRNA	74.9	76.3	0.031	0.020	−0.092	−0.089	91.6
tRNA genes	75.0	78.5	0.091	0.070	−0.373	−0.361	89.6
rrnL gene	74.9	75.4	0.036	0.053	−0.331	−0.360	91.1
rrnS gene	75.2	77.2	0.037	0.026	−0.202	−0.175	86.7

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
