# Peer review of "The First Complete Mitochondrial Genomes of Two Sibling Species from Nitidulid Beetles Pests"

_insects, 2019, doi:10.3390/insects11010024_

Round 1

Reviewer 1 Report

The authors present two mt genomes for closely related Carpophilus dimidiatus and Carpophilus pilosellus. The authors summarize the length and content of each and make comparisons. I think this work will be helpful in future studies of population genetics and other comparative works. 

I only have two minor concerns.

The authors need to have the paper checked for English grammar and verb tense, work order, etc.  The codon families in Fig. 2 are very difficult to read. It would be helpful to include the information in the legend as well as in the figure.

Author Response

Dear reviewer,

Thank you for your comments. All the suggestions are very important and necessary for our MS. Now we are turning to detailed respond to each comment.

Q1:The authors need to have the paper checked for English grammar and verb tense, work order, etc. 

Response: The English had been improved by MDPI.

Q2:The codon families in Fig. 2 are very difficult to read. It would be helpful to include the information in the legend as well as in the figure.

Response: we have we have revised it according to your suggestion. as the following: The relative synonymous codon usage (RSCU) was analyzed for the two mt genomes (Figure 2). The two species shared the same codon families and have the similar feature of RSCU. The trnS1 codon UCN was most commonly used. Furthermore, UUA (trnL1), UCU (trnS1), CGA (trnR) were the top three codons used in the two mt genomes.

In addition, in order to make the method clearer, we adjusted the paragraph order and supply more information about approach and parameters of analysis software. Please see line 64-83.

Sincerely yours,

Yi Wu

Reviewer 2 Report

The manuscript by Wu et al describes characteristics of the mt genomes of two stored product pests and identifies areas of the genomes that differ between these close relatives.  Overall, the data presented herein are useful for developing diagnostic regions for molecular identification of species and geographic populations and could be used by others in the stored product community.  There are some details below that should be expanded upon before the manuscript can be published.  See below for some line-by-line suggestions. 

Line 15 and intro:  more information is needed about why it is so important to develop the mt genome for these two species.  Are they difficult to distinguish from one another and could mtDNA barcoding be a helpful approach for developing markers for identification?  Will you be using these genomes to develop population markers? 

Line 16:  The English needs to be improved in some places of the manuscript.  For example, the phrase “typical set and arrangement of genes present in the ancestral insect” does not fit well in the context of this sentence.  I think you mean “The mt genomes of…..are circular with total assembly lengths lengths of……”  Gene order and content for both species is similar to what has been observed in other insects and consisted of….”

Line 18:  Compared to one another, C. dim. and C. pil. were similar in organization, GC content, codon usage, tRNA secondary structure.  Not sure what you mean by arrangement pattern.  That seems similar to gene organization.  Consider removing that from the paragraph.

Line 20:  Small differences were noted with regards to….

Line 20:  What do you mean by intergenic sequences….do you mean nucleotide similarity of the intergenic spacer region?  Be more specific.  Same with coding regions and control region, what differences are you referring to?  Nucleotide similarity?

Line 28:  What do you mean by “cluster” of stored product insect?   Do you mean taxonomic lineage?

Line 29:  most of which are distributed in tropical and….

Line 32:  This should be “byproducts.”  Also, please specify what these byproducts are from?  Milling or other processes?

Line 33:  They reduce the quality and quantity of stored products and create conditions

Line 34 and throughout:  Please spell out genus name when it falls at the beginning of the sentence. 

Line 36:  dry fruit storage sites and has a widespread…

Line 41:  Please elaborate a bit more on what on what molecular identification studies have been performed already for these two species. 

Line 41:  not only to study the population structure

Line 44:  are usually single circular chromosomes containing 37 genes

Line 46:  relatively stable

Line 48:  phylogeographic

Line 49:  At the time of this report, no complete mt genomes of……were available…..

Line 51:  What do you mean by analyzing the fragmentation?  

Line 56: from a grain processing facility

Line 59:  three adults.  Were the adults pooled into one DNA extraction?  Or the three insects were extracted and analyzed separately?

Line 60:  following manufacturer’s instructions.

Line 60:  A UV-Vis

Line 63:  More information is needed on the adapters and kits that were used to construct the DNA libraries for sequencing.  You mention that you sequenced 250 x 250 PE reads.  What was your fragment size selected after shearing?  This information is important to include because it impacts your assembly.

Line 65:  Please include location for Barry Genomics Company

Line 66:  More information should be provided about this anchor approach.  It seems to be that you first obtained sequence data from this region using PCR coupled with Sanger sequencing and then used these regions as baits to extend the assembly using the Illumina reads.  There are several publications and software programs available that utilize this approach to mtDNA assembly.  Please include some references in this section to support this method and show why it is superior to the shotgun approach.  Some issues such as coverage and repetitiveness can make assembling mtDNA from shogtun data challenging. 

Line 67:  This statement belongs in the next paragraph.  We applied “map to reference” strategy and mapped all cleaned reads to the “anchor”

Line 68:  Please move the steps regarding read quality processing earlier in the paragraph.  The way it is organized right now, you are mentioning the use of clean reads before you are explaining the methodology used to obtain those.

Line 70:  The PCR amplification volume was 25 ul containing 12.5 ul of 2X

Line 73:  The amplicons were sequenced at

Line 75-77:  Please describe the quality filtering parameters used for Trimmomatic.

Line 80:  More information is needed here.  Was this process done recursively or iteratively (over and over again/multiple times) to get the mt sequence?  This information will be useful for readers.

Line 84:  Please indicate the parameters used to annotate the genes by MITOS (ie, which reference data were chosen for gene finding).  Please also describe your blast parameters and mention whether you used blastp/x/n and also please mention your thresholds (e-value, bit-score, etc).

Line 85:  using tRNA-Scan and ARWEN

Line 86:  If all 22 tRNAs were not found using this software,

Line 97:  remove the word “finally” from this statement.

Line 97:  Using the map to reference strategy with the cox1, rrnS, and rrnL genes as anchors, 56,246 reads were assembled for C. dim and XXXX were assembled for C. pil.

Line 100:  consensus sequence lengths of….

Line 105:  was assembled into one circular contig, similar to many other insect mt genomes.

Line 107:  share the same arrangements:  do you mean share the same gene order?

Line 109:  and the other genes were encoded by the heavy strand.

Figure 1:  Please label A and B for the two different species and indicate which genome corresponds to each species.

Line 122:  and C. pil are

Line 122:  As in mt genomes from other insect species,

Line 124:  Similarly, the mt genomes of both C. dim and C. pil had positive AT-skews and negative GC-skews. 

Line 131:  190 bp

Line 133:  What do you mean when you say the total overlapping regions?  Is this with regards to the intergenic spacer region?  Please be more specific here.

Line 139:  stop codons

Line 142:  the cox2 gene had an incomplete stop codon.

Line 146:  The other genes started with ATT

Line 147:  When you say that T was the stop codon, do you mean that this is an incomplete stop codon (similar to the previous paragraph)?

Line 150:  The trnS1 codon UCN was the most commonly used

Line 159:  Please specify if this feature is found in genomes of both species.

Line 161:  Please revise the structure of this sentence.  Most of the XX stems had 5 bp while the stems for six of the tRNAs had 4 bp and the XX stem of trnT was 6 bp.  Also please specify whether these features were conserved between the two genomes or not.

Line 165:  The two rRNA genes were 1275 and XX for C. dim and XXX and YYY for C. pil.  You need to add the word “and” here a few times.  See the sentence structure above.

Line 172:  What did you compare?  We compared the mt genomes of C. dim and C. pil.

Line 173:  Most of these features were similar between the two species and the gene order and other structural features were largely conserved.

Line 174:  main difference

Line 176:  with a nucleotide similarity? 

Line 176:  For PCGs, percent nucleotide (or protein?) similarity was 86.60%, including 1706 variable positions throughout the mt genome.  nad41 had the highest similarity and cox2 had the lowest similarity. 

Line 178:  The average nucleotide similarity of all of the cox genes was XXXX and the average similarity of the nad genes was XXXXX.

Line 181:  What do you mean when you say they are similar?  Similar in terms of nucleotide identity?  Similar in terms of gene organization?  Be more specific about what features of the mt genomes are similar among these stored product insects.

Line 183: Here, the first two mt genome sequences for species within the genus Carpophila are reported and more data from other species in this genus will be needed for…..

Line 184:  Be more specific about what types of research you need these genomes for?  Molecular evolution?  Resolving systematics or phylogenetics studies?   

Reviewer 3 Report

The authors of the manuscript entitled “The First Complete Mitochondrial Genomes of Two Sibling Species from Nitidulid Beetles Pests”, have done really good job sequencing the mt genome of C. dimidiatus and C. pilosellus. All the lab work done by the authors seems to be well-executed and analyzed. My main concern regarding this manuscript though, is the poor English throughout the manuscript. Therefore, the authors may choose to have their manuscript revised by a competent English speaker. For this reason, I would recommend a minor revision of the manuscript (mainly the language part) by the authors before resubmission to INSECTS.

Author Response

Dear reviewer,

Thank you for your comments and time. About the the English of the manuscript, it had been improved by MDPI.

Sincerely yours,

Yi Wu